# The Human Basal Ganglia Mediate the Interplay between Reactive and Proactive Control of Response through Both Motor Inhibition and Sensory Modulation

**DOI:** 10.3390/brainsci11050560

**Published:** 2021-04-28

**Authors:** Marion Criaud, Jean-Luc Anton, Bruno Nazarian, Marieke Longcamp, Elise Metereau, Philippe Boulinguez, Bénédicte Ballanger

**Affiliations:** 1Institute of Psychiatry Psychology & Neuroscience, Department Child & Adolescent Psychiatry, Kings College London, London SE24 9QR, UK; 2Centre IRM-INT@CERIMED, Institut de Neurosciences de la Timone, CNRS UMR7289 & Aix-Marseille Université, 13005 Marseille, France; Jean-Luc.Anton@univ-amu.fr (J.-L.A.); Bruno.Nazarian@univ-amu.fr (B.N.); 3Laboratoire de Neurosciences Cognitives, CNRS UMR 7291 & Aix-Marseille Université, 13005 Marseille, France; marieke.longcamp@univ-amu.fr; 4Hôpital Neurologique Pierre Wertheimer, Service de Neurologie C, Centre Expert Parkinson, Hospices Civils de Lyon, 69677 Bron, France; elise.metereau@chu-lyon.fr; 5Université de Lyon, 69622 Lyon, France; philippe.boulinguez@univ-lyon1.fr (P.B.); benedicte.ballanger@cnrs.fr (B.B.); 6Université Claude Bernard Lyon 1, 69100 Villeurbanne, France; 7INSERM, U 1028, Lyon Neuroscience Research Center, 69000 Lyon, France; 8CNRS, UMR 5292, Lyon Neuroscience Research Center, 69000 Lyon, France

**Keywords:** go/nogo, fMRI, task setting, functional connectivity (PPI), response inhibition, visual attention

## Abstract

The basal ganglia (BG) have long been known for contributing to the regulation of motor behaviour by means of a complex interplay between tonic and phasic inhibitory mechanisms. However, after having focused for a long time on phasic reactive mechanisms, it is only recently that psychological research in healthy humans has modelled tonic proactive mechanisms of control. Mutual calibration between anatomo-functional and psychological models is still needed to better understand the unclear role of the BG in the interplay between proactive and reactive mechanisms of control. Here, we implemented an event-related fMRI design allowing proper analysis of both the brain activity preceding the target-stimulus and the brain activity induced by the target-stimulus during a simple go/nogo task, with a particular interest in the ambiguous role of the basal ganglia. Post-stimulus activity was evoked in the left dorsal striatum, the subthalamus nucleus and internal globus pallidus by any stimulus when the situation was unpredictable, pinpointing its involvement in reactive, non-selective inhibitory mechanisms when action restraint is required. Pre-stimulus activity was detected in the ventral, not the dorsal, striatum, when the situation was unpredictable, and was associated with changes in functional connectivity with the early visual, not the motor, cortex. This suggests that the ventral striatum supports modulatory influence over sensory processing during proactive control.

## 1. Introduction

Action control relies on the ability to suppress undesired or inappropriate motor activations. Driven by influential original models assuming that only external signals can start an inhibitory process intended to race against motor activation [1], most studies have focused on the cascade of processes triggered by stimuli instructing to suppress the preparation or the execution of ongoing responses [2,3]. Here, these mechanisms are referred to as reactive inhibitory mechanisms. There are two classes of reactive inhibitory mechanisms [4]. Some operate by suppressing specific motor activation, leading to selective inhibition of a particular response [5,6,7,8] while others operate by suppressing the signal that triggers the movement, leading to global (non-selective) inhibition of any possible response [4,9,10]. These two forms of reactive control are complementary for the executive system; while selective inhibition allows immediate execution of concurrent responses, non-selective inhibition suppresses any movement and postpones response selection and/or execution. These complementary mechanisms are variously involved in the different behavioural paradigms that have been documented in the literature.

More recently, processes that are implemented in anticipation of stimulus occurrence, referred to as proactive inhibitory mechanisms, have attracted attention [11]. Proactive inhibitory control of response has originally been conceptualized as a gating mechanism that operates by suppressing movement initiation processes, leading to global inhibition of any possible response [12,13]. These non-selective mechanisms would be implemented by default when the environment is uncertain (action restraint), meaning before any stimulus is presented [12,13,14,15,16,17,18,19,20,21,22,23,24,25,26,27]. Proactive inhibitory control of response may also take other forms, like the gating of response selection until sufficient information has been collected to select the correct option [28,29], the modulation of response threshold [5,22,23], the modulation of the degree of implementation of subsequent reactive control [30,31], or an anticipated preparation of the neural inhibitory processes that will be requested by upcoming events [18,24,27,32,33,34,35]. Like reactive inhibitory control, proactive inhibitory control can be selective or not in the sense it can be specifically directed at one particular response or not. This concept guides investigations toward the analysis of pre-stimulus activity [11,36].

There are a number of major issues about the interplay between proactive control and reactive inhibition that remain to be settled. This is mainly due to the fact that acting and stopping represent interacting subfunctions composed of multiple processes that operate concurrently during the entire course of preparation and execution of a movement and are therefore difficult to disentangle [37]. This difficulty is exacerbated by the fact that these subfunctions rely on or appeal to an intricate network of overlapping brain regions integrating online evaluations of the pros and cons of an action in order to stop the overall process at any stage (ibid). Within this framework, the BG obviously play a critical role thanks to their central position within the motor, associative and limbic integrative and interacting circuits (Figure 1) [38,39,40]. Here, we will focus more specifically on non-selective reactive inhibition of response, which is context dependent and thus requires an inhibitory set [13,20,30,31,41] under supervisory predictive control [3,42,43]. However, how this interplay is implemented in the BG and how it operates within functional circuits is still obscure. Proactive non-selective inhibitory control of response may theoretically act by impeding motor activity or by suppressing the signal that triggers the response in advance of stimulation [12,13,21]. However, proactive non-selective inhibitory control of response may also consist of adjusting in advance the sensitivity of the local self-inhibitory circuitry supporting automatic, reactive non-selective inhibition [10]. The two hypotheses are not mutually exclusive. In addition, since an inhibitory set basically relies on expectation and identification of irrelevant stimuli, it likely recruits more cognitive operations than just motor inhibitory processes and makes the interpretation of brain activation hazardous [3,7,36,43,44,45,46,47,48].

Finally, the neural bases of the different forms of inhibitory control are still unclear also because the psychological and anatomo-functional models have evolved separately to a certain extent. Movement preparation and inhibition rely on modular loops through which cortical inputs enter the BG and are forwarded to the thalamus and project back to the cerebral cortex [49,50,51,52]. These different pathways exert various antagonist effects on motor cortical activity. The direct cortico–striato–pallidal pathway relays activating signals from the motor cortex which ultimately amplify motor cortical activity. The indirect cortico–striato–pallido–STN–pallidal pathway has opposite effects which reinforce the default inhibitory outflow of the BG and reduce activity in the motor cortex. Finally, the hyperdirect pathway relays direct cortical signals from motor and non-motor areas to the pallidum via the STN, allowing fast changes in ongoing motor activity. Since efficient movement control is known to depend on balanced activities between the direct excitatory and indirect inhibitory pathways [53], the indirect pathway might support proactive inhibitory control [40]. The hyperdirect pathway is a good candidate for mediating fast reactive inhibition [54]. However, it could also modulate the balance between facilitation and attenuation of motor cortical activity and contribute to set the mode of automaticity of sensorimotor processing [11,55,56]. Finally, the STN may also play a substantial role in proactive inhibition [57,58], either indirectly through the indirect pathway or more directly by relaying a fast signal switching the executive state (i.e., the balance between facilitation and suppression states from direct and indirect pathways) as a function of perceived contextual changes. By establishing a causal link between the experimental manipulation and the behavioral outcomes, studies using deep brain stimulation (DBS) of the STN have been very useful to pinpoint the involvement of the STN in both reactive [59,60,61,62] and proactive [63,64,65,66,67] inhibition of upper limb movements. However, understanding what precisely is modulated in reactive and proactive control of response needs a stronger neurocognitive footing, with particular concern about the various processes possibly involved and intermixed in the two multifaceted functions [3,11,21,68]. In other words, the anatomo-functional bases of the cortico–subthalamic system, and in particular the roles of the two input structures of the BG—the striatum and the STN in response inhibition are still unclear [69,70,71] (Figure 1).

**Figure 1 brainsci-11-00560-f001:**
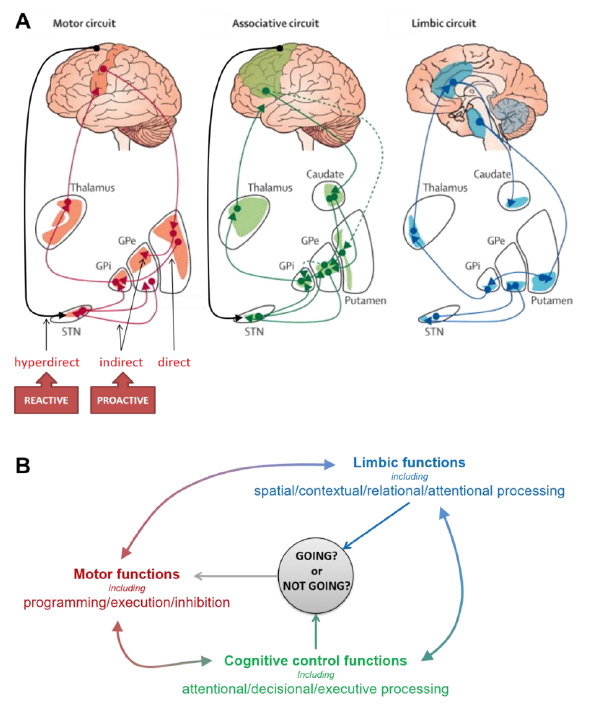
Main functional subdivisions of the cortico–striatal circuits (**A**) and associated functions (**B**) possibly involved in proactive and reactive inhibitory control. The striatum and the subthalamic nucleus (STN), the two input stations of the basal ganglia (BG) receive projections from a range of cortical and subcortical areas. The BG are connected to motor cortical areas (including in particular the motor cortex—M1—the supplementary motor cortex—SMA—the cingulate motor area—CMA) through different pathways that form the motor circuit. It is generally accepted that: (i) the direct pathway, by relaying signals from the motor cortex, activates motor responses, (ii) the indirect pathway, by mediating the default inhibitory outflow of the BG and reducing motor activity, drives proactive inhibitory control, and (iii) the hyperdirect pathway, by relaying direct cortical signals from the SMA to the STN, mediates fast reactive inhibition of ongoing motor responses [72]. However, numerous observations suggest a more complex organization for this multifaceted function [7,11,21]. Furthermore, the BG also have reciprocal connections with non-motor areas forming associative and limbic circuits. These circuits support a wide range of cognitive and limbic functions potentially involved in the control of response inhibition, like attention, action selection, conflict resolution, etc. [73,74]. Each of the BG nuclei is organized topologically through motor, associative and limbic territories that are partially overlapping, allowing for some integration of signals [72]. It is therefore likely that the BG mediate the interplay between reactive and proactive inhibitory control of response by integrating dynamic signals that contribute to decide any time whether to go or not [37]. Adapted from The Lancet Neurology, Rodriguez-Oroz, M.C., Jahanshahi, M., Krack, P., Litvan, I., Macias, R., Bezard, E. and Obeso, J.A., Initial clinical manifestations of Parkinson’s disease: features and pathophysiological mechanisms, 1128–1139, [75] Copyright (2009), with permission from Elsevier.

In this fMRI study we use an anatomical region of interest (ROI) approach in an adapted go/nogo task that emphasizes non-selective inhibition and allows proper recording of both proactive and reactive activity [20,36]. We use functional connectivity to clarify the respective roles of the two input structures of the BG and disentangle their respective effects on cortical activity.

## 2. Materials and Methods

We used a simple go/nogo task intended to minimize the potential confounds often encountered in more complex designs [36]. The present study uses the dataset already presented in [21]. Only the core methodology is presented here (Figure 2). The reader is referred to [21] for more detailed description of the methods. In this previous paper, we performed whole brain contrasts intended to reveal proactive and reactive, selective and non-selective, inhibitory processes with a single event-related fMRI design. Here, we use an ROI approach to focus on the activity of the BG and perform psychophysiological interaction (PPI) analyses to assess whether there is an interaction between the inhibitory control state and the functional coupling between the BG and the rest of the brain.

### 2.1. Research Participants

Twenty right-handed participants (20 to 42 years old, 10 males) without history of psychiatric or neurological disease were included in the study. The experiment was performed in compliance with the code of ethics of the World Medical Association (Declaration of Helsinki) and the protocol was preapproved by the appropriate ethical committee in Biomedical Research (CPP sud-est IV, N°11/025). All participants gave written informed consent and were paid EUR 50 for their participation.

### 2.2. Go/Nogo Task, Apparatus and fMRI Design

Lying down in the scanner, subjects held in the right hand an amagnetic handle positioned below the sternum and equipped with a highly sensitive button. They were looking at a visual display composed of a panel equipped with light-emitting diodes through a mirror. Participants were asked to react as fast as possible to go stimuli while refraining from reacting to nogo stimuli (Figure 2A) by pressing—or refraining from pressing—the button with the right thumb.

Importantly, to limit the cognitive load of the task and avoid confusion with other cognitive processes frequently solicited in complex designs, we used: (i) a simple stimulation design conveying only one bit of information to reduce the complexity of identification of signals, (ii) stimulus–response associations that remain stable throughout the task to avoid switching, (iii) equiprobable go and no-go signals to avoid biasing attention, (iv) a single button-pressed response with the right thumb rather than a choice between hands or digits or a goal-directed movement, (v) a triple training session and a feedback-based procedure to control that RT fell within the calibrated distribution of prepotent responses (see detailed procedure in [21], Section 2.2). It has been repeatedly observed that simple designs involving a simple speeded reaction time task with a single response and only one bit of information are ideal to involve prepotent responses and to elicit covert automatic motor activations that must be counteracted by inhibitory mechanisms [14,15,76,77]. To this respect, ensuring that responses are prepotent is essential to guarantee the need for inhibition. In the same vein, this kind of simple design associated with an online control of RT distribution is more likely to limit the context effect [78] often observed in complex designs using elaborated responses, which consists of completing the movement plan after movement onset rather than inhibiting the motor command completed during the RT period.

At the beginning of each trial, the fixation point could turn either green or red. The green cue indicated that no nogo stimulus would be presented (go_control_trials only). In this control condition, participants were free to react automatically to any upcoming event (i.e., no proactive inhibition was required). By contrast, the red cue indicated that a go stimulus (go_trials), a nogo stimulus (nogo_trials) or no stimulus at all (catch trials) could occur. In this uncertain context, participants were invited to react as fast as possible to go stimuli while refraining from reacting to nogo stimuli.

Stimuli presentation and behavioral data acquisition were performed with software specifically developed using the LabVIEW^®^ IDE, National Instruments Corp., Austin, TX, USA. Pre-stimulus delays (time between the beginning of a trial and stimulus presentation) varied randomly from two to six seconds (in steps of 1 s to make stimulus occurrence unpredictable. The experiment was composed of four acquisition runs. Each run included 20 go trials, 20 nogo trials, 20 go_control trials and 20 catch trials randomly presented. The experiment was composed of 320 trials in total.

### 2.3. Data Acquisition

Images were acquired on a 3-T MEDSPEC 30/80 AVANCE whole-body imager (Bruker, Ettlingen, Germany), equipped with a circular polarized head coil. A high-resolution structural T1-weighted image (MPRAGE sequence, resolution 1 × 0.75 × 1.22 mm) in sagittal orientation, covering the whole brain, was acquired for each subject. A T2*- weighted echoplanar sequence, covering the whole brain with 28 interleaved 3-mm- thick/0-mm-gap axial slices (repetition time = 1867 ms, echo time = 30 ms, flip angle = 77°, field of view = 19.2 × 19.2 cm, 64 × 64 matrix of 3 × 3 mm voxels) was used for functional imaging. A number of 337 functional volumes per session were acquired during four sessions, for a total of 1348 volumes per participant.

### 2.4. Data Analysis

#### 2.4.1. Behavioral Analysis

The false alarm rate and the mode of individual distributions of reaction time (RT) were used as the main behavioral dependent variables to better characterize errors and fast prepotent responses, respectively, in group analyses [21]. As the lengthening of reaction time (RT) in the red cue condition with respect to the green cue condition provides a reliable marker of the involvement of proactive inhibitory control [12,13,14,20,21], the difference in RT between these two conditions (delta RT) was also used to further assess brain–behavior correlations after PPI analyses. Delta RT (i.e., proactive slowing) and functional connectivity changes (beta values) were correlated across subjects when appropriate.

#### 2.4.2. fMRI Preprocessing

Data were processed using SPM8 software (http://www.fil.ion.ucl.ac.uk/spm/, Wellcome Department of Imaging Neuroscience, University College London, UK, 2 March 2015), according to the general linear model (GLM—[79]). The first six functional volumes of each run were removed to eliminate non-equilibrium effects of magnetization. The remaining 331 images were corrected for differences in slice acquisition time. The images were then corrected for head movements by realigning all the images with the first image using rigid body transformations and unwrapped according to the fieldmap recording. Spatial normalization was improved using the DARTEL toolbox on an MNI template. Data were spatially smoothed with an isotropic Gaussian filter (8 mm FWHM).

#### 2.4.3. Regions of Interest (ROI)

We used an anatomical ROI approach to focus on the respective activity of the BG, defined at the group level. Fourteen regions were defined with the ATAG atlas [80]: left and right subthalamic nucleus (STN), globus pallidus internal segment (GPi), globus pallidus external segment (GPe), substantia nigra (SN), and the red nucleus. Due to the difficulties to segment the ventral and dorsal striatum, we used the fsl-oxford-striatal-atlas. In the atlas the dorsal boundary of the ventral striatum was defined included the medial part of the caudate and the rostroventral part of the putamen (fsl-oxford-striatal-atlas; [81]).

#### 2.4.4. Event-Related Analyses of BOLD Signal Changes

In the first level of the statistical analysis, 16 event types were defined: 10 effects of interest (5 types of trial—go_control, go, nogo, catch_control, catch_nogo—for 2 periods—pre-stimulus and post-stimulus) as well as 6 effects of no interest (short pre-stimulus delays -2 to 3 s- and inter-trial interval). The events were time-locked to the onset of the stimulus (post-stimulus) or to the onset of the cue (pre-stimulus), modelled according to their onset and their duration, and convolved with a canonical HRF (Figure 2). Data were high pass-filtered at 128 s and summarized into three contrasts performed for each participant [21]. One sample *t*-tests were used to assess:-Reactive, selective, inhibitory mechanisms: the difference in stimulus evoked activity between the nogo and go conditions (contrast ([nogo]-[go])).-Reactive, non-selective, inhibitory mechanisms: the difference in stimulus evoked activity between the two conditions of uncertainty (contrast ([nogo + go]-[go_control])). The contrast was balanced by weighting the go_control condition (x2) to compensate for the unequal number of trials.-Proactive, non-selective, inhibitory mechanisms: the difference in activity implemented in the pre-stimulus period between the two conditions of uncertainty (contrast [(red cue)-(green cue)]).

The SPM group maps were generated with random-effects models. The resulting individual statistical maps were entered into one-sample *t*-tests. We used a corrected cluster level of *p* < 0.01 by applying a voxel-level threshold of *p* < 0.001 (uncorrected for multiple comparisons) and a cluster extent of more than 50 contiguous voxels.

#### 2.4.5. Psychophysiological Interactions (PPI)

When significant activations were found in the BG, context-dependent changes in functional connectivity were assessed using PPI analyses [82,83]. The anatomical ROI served as seed regions for extracting the eigenvariate of the fMRI signal. For each participant and each ROI, the PPI variable including the PPI term was calculated for each one of the three contrast matrices described above. These terms were used to model a GLM for each participant. At the group level, contrast images of the PPIs were analyzed using one-sample *t*-tests. We used a corrected cluster level of *p* < 0.01 by applying a voxel-level threshold of *p* < 0.001 (uncorrected for multiple comparisons) and a cluster extent of more than 50 contiguous voxels.

## 3. Results

### 3.1. Behaviour

The commission error rate was low (10.4 ± 5.8% of nogo trials) indicating good overall inhibitory performance, an essential prerequisite for undertaking a proper analysis of RT data. A main effect of Uncertainty (F(1, 19) = 142.87, *p* < 0.001) showed significantly longer go RT (384 ± 48 ms) than go_control RT (305 ± 55 ms).

### 3.2. Event-Related Analyses of BOLD Signal Changes

#### 3.2.1. Reactive, Selective Brain Activity

ROI analyses returned no significant BOLD changes.

#### 3.2.2. Reactive, Non-Selective (Context Dependent) Brain Activity

ROI analyses revealed non-selective response to stimuli when presented in an uncertain context in the left dorsal striatum—dorsal putamen—((−27, 9, 6), z-score: 4.51; cluster size: 25 voxels), the left STN ((−15, −15, 0), z-score: 4.54; cluster size: 11 voxels), the left GPi ((−18, −12, 0), z-score: 4.10; cluster size: 8 voxels, Figure 3). PPI analyses returned no significant changes in functional connectivity induced by the experimental conditions.

#### 3.2.3. Proactive, Non-Selective (Context Dependent) Brain Activity

ROI analyses revealed activity associated with proactive control in the right ventral striatum ((12, 18, −6), z-score: 3.54; cluster size: 24 voxels, Figure 2). PPI analyses using the ventral striatum as seed region revealed significant increase in functional connectivity with the visual cortex (right Lingual gyrus extending to left (BA 18, [15, −81, 3]; z-score: 3.85; cluster size: 128 voxels)) induced by proactive control. A positive correlation (rho = 0.405, *p* = 0.038) was found between proactive slowing (delta RT) and functional connectivity changes between the ventral striatum and visual areas (beta values), indicating that the stronger the changes in functional connectivity between these areas, the larger the behavioral effect (Figure 3).

## 4. Discussion

Thus far, the striatum and the STN have been variously associated with different inhibitory processes, selective and non-selective, reactive and proactive [11,13,27,31,54,55,57,60,66,84,85,86]. However, the exact scenarios in which the striatum and the STN are engaged in response inhibition have been debated for many years because it is difficult to disentangle the different processes possibly involved in standard inhibitory tasks [36]. In the present study, we confirm the involvement of the dorsal striatum, the STN and the GPi in reactive, non-selective inhibition. However, we also show that the striatum, through its ventral part, plays a complementary and efficient role in anticipation of stimulus occurrence in uncertain contexts by modulating activity in the visual cortex.

### 4.1. Advantages and Limitations of the Experimental Task

Before interpreting these main results, it must be emphasized that the task used in the present study has been especially designed to reduce the risk for potential confounds with a number of cognitive functions often involved substantially in more complex designs [36,45,46,48]. This is a prerequisite for clarifying the role of the BG in response inhibition. However, it is likely that such a reductionist procedure has lowered the involvement of selective inhibitory mechanisms [21]. Obviously, the fact that no activity associated with reactive selective inhibition has been evidenced in the present data will not be used to invalidate selective models.

### 4.2. The Role of the Dorsal Striatum, STN and GPi in Reactive Non-Selective Inhibition

As the two input stations of the BG and the output nucleus of the BG, respectively, it is not surprising to identify the dorsal striatum, the STN and the GPi in reactive inhibition (Figure 3). While the putamen is commonly reported in fMRI designs testing reactive response inhibition [7], direct clear evidence for the STN and the GPi is still sparse. One of the main difficulties in understanding the various possible functions of these structures is that their small size, deep location and anatomical complexity make it challenging to acquire reliable signal [87]. This difficulty is amplified when the behavioral task is complex and recruits too many concurrent functions. In the simplified version of the go/nogo task used in the present study, we provide direct evidence for the involvement of the STN and GPi in reactive, non-selective (also called global) inhibition. This result was very much expected because surgical interventions such as deep brain stimulation in Parkinson’s disease (PD) have been much helpful to demonstrate the involvement of the STN in motor and inhibitory control [40,55,62,88]. However, the role of the GPi inferred from this class of studies is less clear. It would facilitate action initiation in the ‘Go’ pathway, but GPi-DBS would not alter the functioning of the indirect ‘NoGo’ pathway [89]. Accordingly, because it did not allow the highlighting of the pattern of functional connectivity of BG reactive activity, the present dataset raises more questions than it solves. Functional MRI experiments specifically designed for using dynamic causal modelling might be fruitful, as [47,90] recently did for providing evidence for the role of the STN in reactive inhibitory control through the hyperdirect pathway. Yet, a comprehensive overview including all BG structures of interest and all modes of control is still missing. It is likely that combining DBS with high temporal resolution electrophysiological recordings of specific BG targets and large-scale cortical activity also provides greater chances to identify the dynamics supporting the interplay between cortical and subcortical activity in the different modes of inhibitory control. Very promising data have recently been provided by Hell and colleagues [91], who clearly showed that the STN implements a dynamic executive control signal regulating the interplay between proactive and reactive mechanisms. However, it is still challenging for electrophysiological methods to record simultaneously all BG structures of interest and to localize the cortical sources involved in the interactions of interest.

Other issues are questionable in the present data. Why the dorsal striatum, STN and GPi show increased activity unilaterally is not clear. Although this left lateralization pattern is usually associated with right-hand response execution, previous observations contradict it. It has indeed been shown that bilateral [59,63] but not unilateral DBS [92] restores reactive inhibitory control to a near-normal level in PD patients. Consistent with DBS studies, bilateral activation has been found in the dorsal striatum, STN and GPi in classical stop signal reaction time tasks [93]. Yet, studies using different tasks have also found different patterns. Recently, Maizey and colleagues [94] used an even more complex design combining contextual inhibition with the solicitation of multiple responses in choice RT tasks, and observed a right lateralization of activity for all BG structures. There is no consensus that accounts for these differences. Nevertheless, this pattern can be related to the activation of the right lateralized parieto–frontal cortical network frequently observed in response inhibition tasks. Much of this right cortical activity has been attributed to more general cognitive processes more or less recruited as a function of task complexity (which can be objectively estimated on the basis of its requirements for working memory, response selection, rules switching and attention:, e.g., [36]). In other words, as choice of task plays a major role in the inhibitory and non-inhibitory processes at stake, it is tempting to speculate that the left pattern of BG activity issued from our simple task—intended to suppress classical confounds—is specific to global, non-selective response inhibition of right hand movements, while the involvement of processes contributing to select one response among alternatives would engage more right lateralized circuits. Obviously, a huge amount of work is still needed to disambiguate the issue of lateralization of response inhibition, far beyond the BG [7,36,95,96,97].

### 4.3. The Role of the Ventral Striatum in Proactive Inhibition

Surprisingly, we did not find significant activity in the dorsal striatum or the STN during proactive control while a major open issue in the field is not whether but rather how these regions are involved in the function. By contrast with reactive control that has been most commonly associated with the STN, proactive control has been most commonly associated with striatal areas [13,33,35,84,86,98,99,100,101]. The dominant view is that, while reactive control is exerted through the hyperdirect pathway, proactive control is exerted through a competition of facilitation and suppression signals within the direct and indirect pathways. Yet, there are converging clues from human studies using direct electrophysiological recordings [57,58,71] or studies combining DBS and cortical activity recordings [91,101] that the STN mediates proactive control and the switch between control states, possibly via the indirect and hyperdirect pathways, respectively [64]. Although the present data provide null results, we do not reject this hypothesis. There are good reasons, beyond the technical limitations evoked above [87], that our contrasts simply missed the activity of interest. Since BG regions are involved in motor execution as well as response inhibition, it is tricky to disentangle concurrent activations (either excitatory or inhibitory) with an indirect marker like blood flow [99,102,103,104].

The major finding of the present study is that the ventral striatum plays a substantial role in proactive inhibitory control (Figure 3). This is uncommon given that most human and rodent studies suggest that the nucleus accumbens (NAc) does not participate directly in motor response inhibition [101,105,106]. It is generally admitted that the NAc is more likely involved in cognitive and limbic, not motor functions [107,108,109,110]. Still, it would mediate “waiting impulsivity” [111,112], a construct referring to the processes involved in action restraint during the selection between alternative responses when reward is expected. However, our simple detection task does not involve reward discounting or decision between various possible responses. Thus, the activation of the ventral striatum observed here is unlikely related to motor or reward functions. The PPI analyses rather put forward the role of the ventral striatum in the modulation of early visual processing when the situation is unpredictable and requires enhanced visual attention during the prestimulus period. There is already strong evidence that attentional biasing signals in human visual cortex can be obtained in the absence of any visual stimulation, when subjects are just expecting visual stimuli [94]. In response inhibition experiments, distinguishing between appropriate and inappropriate stimuli is a prerequisite for providing appropriate responses to each type of stimulus, and attentional and inhibitory processes are obviously intricately linked, if not confounded [3,42,43,45,46,47,48]. A major finding of the present study is that the human ventral striatum likely plays a major role in this function, contributing indirectly to inhibitory performance as the stronger the changes in functional connectivity between the ventral striatum and visual areas, the larger the behavioral effect (Figure 3). This conclusion is consistent with animal studies suggesting that, while limited evidence supports the direct involvement of the ventral striatum in response inhibition, it is involved in attentional processes associated with the functions of set shifting, switching and monitoring that are essential for inhibitory control in changing and uncertain environments [74,113] but often overlooked [7,36,114]. It is also consistent with the known pattern of structural connectivity of the NAc, which constitutes the striatal way station of the limbic circuit integrating thalamic, hippocampal, amygdaloid and medial prefrontal signals, which reaches pallidal, subthalamic and substantia nigra targets, but which is not directly connected to the visual cortex [115].

There are still, however, open issues in the present data. Why the NAc unilaterally shows increased activity while the visual cortex shows increased functional connectivity bilaterally is not clear. Broadly, NAc anatomical and functional resting state connectivity patterns are characterized by high inter-hemispheric symmetry [115]. Yet, functional data have reported interesting findings about asymmetrical patterns of functional connectivity between the left/right NAc and the occipital cortex in disorders associated with dysfunctional mesolimbic reward processing and response control [116]. Broadly, right NAc connectivity patterns seem to be more varied, and more altered than left NAcc patterns in clinical conditions (ibid). No clear interpretation has been provided by the authors, but it must be noticed that, in the same paper, a double trend-level relationship has been found between right NAc connectivity and impulsivity on the one hand, and left NAc connectivity and sensation seeking on the other hand [117]. Taken together, these two studies raise the issue of asymmetrical accumbens connectivity with attention/motor/default networks and reward circuitry. Further work is needed to address these specific questions.

More generally, our observations support the view that the NAc does not only serve as a reward center but also plays a key role in various cognitive functions contributing to action control and the suppression of inappropriate actions, including visual attention [74]. This conclusion is consistent with studies suggesting that the ventral striatum plays a substantial role in the identification and learning of associations between visual stimuli in general, not simply during reward learning [118]. It is also in line with previous empirical evidence that the basal ganglia are broadly implicated in attentional flexibility [119,120,121], with the ventral striatum playing the pivotal role in the gating function in the domain of attention [122]. By supporting modulatory influences on action performance when the course of action is ambiguous or uncertain [74], the ventral striatum likely has a key role in the interplay between proactive and reactive control of response.

## 5. Conclusions

Inhibitory control requires a combination of both reactive and proactive signals in BG loops. Yet, the interplay between reactive and proactive mechanisms of control is not only an issue concerning the modulation of motor processing in the direct, indirect and hyperdirect circuits. It also has to do with attentional supervisory control and the modulation of sensory processing through other BG loops involving the ventral striatum. Understanding how the distinct circuits within and outside the striatum interact to coordinate cognitive and motor control remains a central issue [3,42,43,123] with major implications for the interpretation and clinical management of patients with cognitive and motor dysfunctions related to BG disorders [118,124]. More precisely, multiple and opposite symptoms can result from disorders of various inhibitory functions depending on distinct cortico–striatal substrates. For instance, it is widely accepted that different forms of reactive and proactive inhibition dysfunction can lead to different forms of impulsivity in various clinical conditions [111,125,126,127,128], but it is less known that dysfunction of proactive control, by implementing exaggerated inhibition, can also lead to akinesia and related motor depletion [129,130]. It is a major issue to identify more precisely the different circuits, mechanisms and neurotransmitter systems that contribute to inhibitory control, and how these intricate functional loops interact (e.g., [40,64,126,131,132]), to find therapeutic answers to symptoms without current satisfactory solutions.

## Figures and Tables

**Figure 2 brainsci-11-00560-f002:**
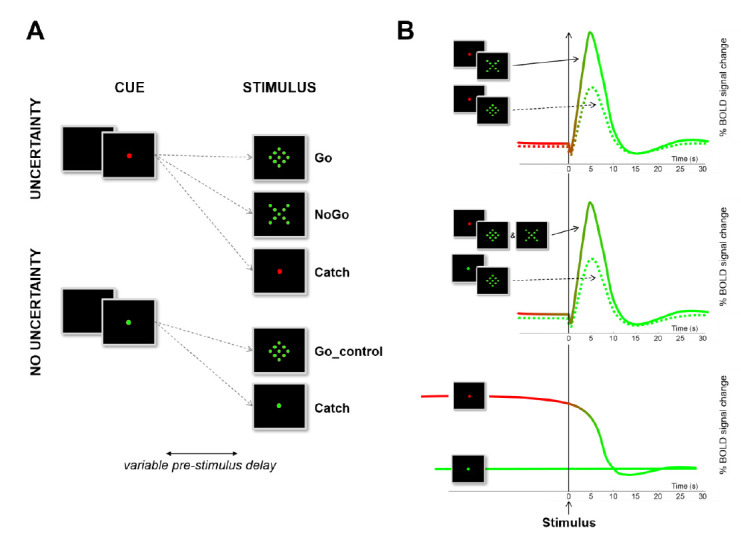
Design and rationale. (**A**) Illustration of the design used to assess three different forms of inhibitory control with a simple Go/NoGo task. Participants were asked to react as fast as possible to a Go stimulus (diamond) by means of a button press, or to withhold the prepotent response to an equiprobable NoGo stimulus (X). (**B**) Illustration of the contrasts used to identify the different possible forms of response inhibition on the basis of their respective BOLD dynamical signature. The reader is referred to [21] for more details about the methods and to [36] for a critical review and a meta-analysis of the standard contrasts used to evidence reactive inhibition in Go/NoGo tasks.

**Figure 3 brainsci-11-00560-f003:**
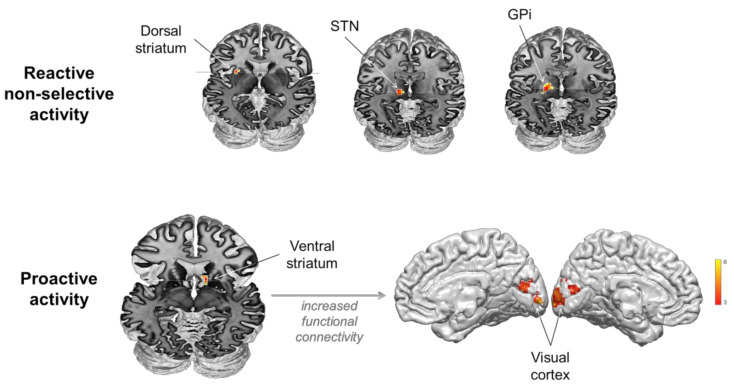
Involvement of the basal ganglia in reactive non-selective and proactive mechanisms during inhibitory control.

## Data Availability

The data presented in this study are available on request from the corresponding author.

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
