# Peer review of "The Human Basal Ganglia Mediate the Interplay between Reactive and Proactive Control of Response through Both Motor Inhibition and Sensory Modulation"

_brainsci, 2021, doi:10.3390/brainsci11050560_

Round 1
Reviewer 1 Report
In the present manuscript, functional magnetic resonance imaging in humans was used to investigate which nuclei of the basal ganglia play a predominant role in the initiation or suppression of motor responses (pressing a button) to a visual stimulus with and without proactive control.
An important finding was that the nucleus accumbens, in particular, plays a role in the suppression and initiation of responses to a visual signal and is associated with the primary visual cortex.
Materials and Methods, refers to a previous publication for a detailed description of the experimental setup. For easier reading and faster interpretation possibility, it would be better if the experimental setup is also described in the current manuscript. For example, with which hand was the button pressed?
It is not addressed in the discussion why the nucleus accumbens unilaterally shows increased activity, but the primary visual cortex shows increased activity bilaterally. What data on the connectome are known here?
In principle, it is useful to visualize the working hypothesis as well as the obtained results about the wiring of the basal ganglia in terms of initiation, inhibition and proactive control of motor responses in the form of one or more graphical schemes.
A list of abbreviations is lacking.
In chapter is 4.2. there are several references to "GB". Was BG meant?
Author Response
REPLY TO REVIEWS brainsci-1178129
We are deeply grateful to the reviewers for their constructive comments and useful suggestions as well as to the editor for giving us a chance to improve our work. We have carefully addressed anyone of the comments. We hope that the changes made in the manuscript have substantially improved the relevance and the outlook of this original article, and that the paper now reaches the high standards of the journal and the expectations of this special issue. These changes are itemized below, and are highlighted in the revised manuscript.
REVIEWER 1
In the present manuscript, functional magnetic resonance imaging in humans was used to investigate which nuclei of the basal ganglia play a predominant role in the initiation or suppression of motor responses (pressing a button) to a visual stimulus with and without proactive control.
An important finding was that the nucleus accumbens, in particular, plays a role in the suppression and initiation of responses to a visual signal and is associated with the primary visual cortex.
- Materials and Methods, refers to a previous publication for a detailed description of the experimental setup. For easier reading and faster interpretation possibility, it would be better if the experimental setup is also described in the current manuscript. For example, with which hand was the button pressed?
We agree with Reviewer 1. More information about the experimental setup has been provided in the Materials and Methods section (P3).
- It is not addressed in the discussion why the nucleus accumbens unilaterally shows increased activity, but the primary visual cortex shows increased activity bilaterally. What data on the connectome are known here?
This is indeed a notable issue and a good idea. Because there are some data available in the literature, we did not use ourselves the connectome to answer these questions. But Xia and collaborators did it recently (Multimodal connectivity-based parcellation reveals a shell-core dichotomy of the human nucleus accumbens, Human Brain Mapping 38:3878–3898, 2017). They extracted experimental data from the Human Connectome Project and used tractography and resting state functional MRI to track the anatomical and functional connectivity of the NAc in healthy adults. Their results confirm the primary connections of the shell and core from earlier animal studies (the NAc receives signals primary from the frontal lobe, hippocampus and amygdala and projects to the ventral pallidum and other subcortical motor effector sites). No evidence was found for connections with the visual cortex, neither anatomical nor functional. Broadly, NAc anatomical and functional resting state connectivity patterns are characterized by high inter-hemispheric symmetry.
However, other functional data have reported interesting findings about asymmetrical patterns of functional connectivity between the left/right NAc and the occipital cortex in disorders associated with dysfunctional mesolimbic reward processing and response control (Cservenka et al., 2014). Broadly, right NAc connectivity patterns seem to be more varied, and more altered than left NAcc patterns in clinical conditions (ibid). No clear interpretation has been provided, but it must be noticed that, in the same paper, a double trend-level relationship has been found between right NAc connectivity and impulsivity and left NAc connectivity and sensation seeking (see also Weiland et al., 2013). Taken together, these two studies raise the issue of asymmetrical accumbens connectivity with attention/motor/default networks and reward circuitry. More generally, these observations support the view that the NAc does not only serve as a reward center but also plays a key role in various cognitive functions contributing to action control and the suppression of inappropriate actions, including visual attention (see Floresco 2015 for review).
These aspects have been discussed in the text in relation to our data (P 17-19).
- In principle, it is useful to visualize the working hypothesis as well as the obtained results about the wiring of the basal ganglia in terms of initiation, inhibition and proactive control of motor responses in the form of one or more graphical schemes.
We agree with Reviewer 1. A new figure (Fig. 1) has been added to visualize the working hypothesis within the global theoretical context.
- A list of abbreviations is lacking. In chapter is 4.2. there are several references to "GB". Was BG meant?
All minor comments have been considered carefully. Thank you for identifying and listing.

Reviewer 2 Report
The paper deals with the topic of proactive/reactive control using a sound methodological approach. Results are fascinating and in line with the Special Issue. I have a few suggestions, mainly aimed to enrich the Introduction and the Discussion. By the way, please, in the revised version, use the line numbers.
Major points
- Page 2 Second paragraph. Another way proactive inhibition could delay movements until sufficient information to select the correct option avoids impulsive decisions acting on the frontal cortex (Cavanagh et al. 2011; Frank 2006). This modality should be acknowledged.
- Page 2 Bottom of the page& Discussion paragraph 4.2. and 4.3 When exploiting the stop-signal task, bilateral deep brain stimulation (DBS) of STN has been repeatedly shown to improve reactive (e.g., Mirabella et al. 2012; van den Wildenberg et al. 2006, Swann et al. 2011, Mirabella et al. et al. 2012; van Wouwe et al. 2017) and proactive inhibition of upper limb movements (Mirabella et al. 2013). Notably, Mirabella et al. 2013 showed that bilateral stimulation of STN restores a planning strategy for reaching arm movement generation appropriate to the context in which Parkinson's patients are embedded, i.e., similar to those of healthy subjects. In Mirabella et al. (2013), proactive inhibition was evaluated by comparing the reaction times (RT) and the movement times (MT) of no-stop trials with those measured during the execution of the same movements in the context of a simple RT-task (go-only trial). In fact, it has been repeatedly shown (Mirabella et al. 2008) that when a subject performs a no-stop trial, its RT is lengthened, and its MT is shortened with respect to when he/she performs a go-only trial. This phenomenon represents an optimization of the motor strategy in the two different contexts, and thus, it has been named 'context effect' (Mirabella et al. 2008). Interestingly, as far as the laterality issue of the inhibitory network is concerned, unilateral STN DBS does not affect reactive or proactive inhibitory control of upper limb movements (Mancini et al., 2019). I suggest discussing this evidence because first, the assessment of the effect of proactive inhibition performed as described above minimize most if not all the confounds typical of more complex designs such as the conditional stop-signal (e.g., Aron et al. 2007) or stop-signal anticipation tasks (e.g., van Hulst et al. 2018). Under these conditions, the load on attentional and working memory is high. Thus, outcome measures cannot be easily ascribed to the effect of proactive strategies because of the concurrent cognitive demands on other executive functions. Second, DBS experiments, differently from recording techniques, allow establishing a causal link between the experimental manipulation and the behavioral outcomes.
- Page 2 Third paragraph I believe that an original interpretation of inhibitory control's neural basis can be found in Mirabella (2014). In that paper, it has been proposed that acting and stopping represent functions emerging from specific interactions between largely overlapping brain regions, whose activity is intimately linked (directly or indirectly) to the evaluations of the pros and cons of an action. Such a mechanism would allow the brain to flexibly perform as different functions could be computed, exploiting the same components operating in various configurations. I suggest considering this hypothesis.
- Page 3 Paragraph 2.2 In what sense you suggestedparticipants 'to exert proactive inhibitory control'? Did you explicitly tell them to slow down the movements? This would represent a critical flaw. Subject automatically and largely unconsciously follow this strategy. If you provide explicit instruction, you can alter the experimental results. Please clarify.
- Page 4 paragraph 2.2.4 I do not completely agree on the rationale of the contrast to uncover the neural underpinnings of a reactive no-selective inhibitory mechanism. In my opinion, you should compare the Go_control (certain condition) just with the Go (uncertain condition). This way, you would compare the same number of trials and two conditions where the visual stimuli are perfectly matched. I would suggest performing such analyses.DISCUSSION. You found that reactive non-selective inhibition is underpinned by three basal ganglia in the left hemisphere, while the right hemisphere processed proactive inhibition. Maybe it would be worthy of discussing your findings in light of the influential idea suggesting that the right hemisphere is more involved in inhibitory control than the left hemisphere (e.g., Aron et al. 2014). This is a hotly debated question as several studies showed the opposite, i.e., they provide evidence showing that the inhibitory network subserving upper limb movements is distributed in both hemispheres (Klein et al., 2016). For instance, it has been shown that the left inferior frontal gyrus has a key role in inhibitory control (Swick et al. 2008). Furthermore, it has also been shown that bilateral (Mirabella et al. 2012; Mirabella et al. 2013) but not unilateral DBS (Mancini et al. 2019) restores reactive inhibitory control to a near-normal level. Besides, Mirabella et al. (2017) compared the inhibitory performance of right- and left-dominant Parkinson's patients (RPD and LPD, respectively) in the middle stage of the disease (H&Y-2 or -3), and they did not find any difference in either reactive or proactive inhibition between LPD and RPD patients, even though patients were impaired with respect to healthy controls. This finding has recently been confirmed, testing PD patients in the early stage of the disease when the disease is unilateral (Di Caprio et al. 2020). All in all, this evidence suggests that inhibitory control does not rely solely on the right hemisphere but on the cooperation between the two hemispheres. How do you frame your findings with respect to this topic?
- CONCLUSIONSI would suggest including another consideration, i.e., understanding the interplay between proactive and reactive inhibition is a crucial step towards understanding disorder characterized by poor urge control (e.g., for a recent review, see Mirabella 2021).
Minor points
- ABSTRACTI can't entirely agree with the fact that the role of BG in inhibitory control has been known 'essentially from clinical studies.
- Page 3 Top of the pageCorrect 'ST' to 'STN.'
- Figure 1Why do you expect that bold activity during No-Go trials should be higher than in Go trials? Is such expectation grounded on actual data? Analogously, the other two panels reflect a working hypothesis about how the activity could change in the two conditions or are they simply examples? Please clarify
- RESULTS Paragraph 3.2.3Remove the bold style.
References
- Aron AR, Behrens TE, Smith S, Frank MJ, Poldrack RA. Triangulating a cognitive control network using diffusion-weighted magnetic resonance imaging (MRI) and functional MRI. J Neurosci 2007; 27: 3743–52.
- Cavanagh et al (2011) Subthalamic nucleus stimulation reverses mediofrontal influence over decision threshold. Nat Neurosci.14:1462-7.
- Di Caprio et al. (2020) Early-stage Parkinson's patients show selective impairment in reactive but not proactive inhibition. Mov Disord. 35:409-418
- Frank Hold your horses: a dynamic computational role for the subthalamic nucleus in decision making. Neural Netw. 2006 Oct;19(8):1120-36
- Klein, P.A.; Duque, J.; Labruna, L.; Ivry, R.B. Comparison of the Two Cerebral Hemispheres in Inhibitory Processes Operative during Movement Preparation. NeuroImage 2016, 125, 220–232
- Mancini et al (2019) Unilateral Stimulation of Subthalamic Nucleus Does Not Affect Inhibitory Control. Front. Neurol. 9:1149
- Mirabella et al (2008). Context influences on the preparation and execution of reaching movements. Cognitive Neuropsychology. 25:996-1010
- Mirabella et al (2012) Deep Brain Stimulation of Subthalamic Nuclei Affects Arm Response Inhibition In Parkinson's Patients. Cereb Cortex 22:1124-323.
- Mirabella et al (2013) Stimulation of subthalamic nuclei restores a near normal planning strategy in Parkinson's patients. PLoS One. 8(5):e62793.
- Mirabella et al (2017) Inhibitory control is not lateralized in Parkinson’s patients. Neuropsychologia. 102:177-189
- Mirabella G. (2021) Inhibitory control and impulsive responses in neurodevelopmental disorders. Dev Med Child Neurol. 63 (5): 520-526
- Swann et al (2011) Deep brain stimulation of the subthalamic nucleus alters the cortical profile of response inhibition in the beta frequency band: a scalp EEG study in Parkinson's disease. J Neurosci 31(15): 5721-5729
- Swick, D., Ashley, V., and Turken, A.U. (2008). Left inferior frontal gyrus is critical for response inhibition. BMC Neurosci 9, 102
- van den Wildenberg et al. (2006) Stimulation of the subthalamic region facilitates the selection and inhibition of motor responses in Parkinson's disease. J Cogn Neurosci.18:626-36
- van Hulst BM, de Zeeuw P, Vlaskamp C, Rijks Y, Zandbelt BB, Durston S. Children with ADHD symptoms show deficits in reactive but not proactive inhibition, irrespective of their formal diagnosis. Psychol Med 2018; 48: 2515–21
- van Wouwe et al (2017). Focused stimulation of dorsal subthalamic nucleus improves reactive inhibitory control of action impulses. Neuropsychologia 99: 37-47

Author Response
REPLY TO REVIEWS brainsci-1178129
We are deeply grateful to the reviewers for their constructive comments and useful suggestions as well as to the editor for giving us a chance to improve our work. We have carefully addressed anyone of the comments. We hope that the changes made in the manuscript have substantially improved the relevance and the outlook of this original article, and that the paper now reaches the high standards of the journal and the expectations of this special issue. These changes are itemized below, and are highlighted in the revised manuscript.
REVIEWER 2
The paper deals with the topic of proactive/reactive control using a sound methodological approach. Results are fascinating and in line with the Special Issue. I have a few suggestions, mainly aimed to enrich the Introduction and the Discussion. By the way, please, in the revised version, use the line numbers.
Major points
- Page 2 Second paragraph. Another way proactive inhibition could delay movements until sufficient information to select the correct option avoids impulsive decisions acting on the frontal cortex (Cavanagh et al. 2011; Frank 2006). This modality should be acknowledged.
We agree with Reviewer 2. We have clarified this point in our initial reference to these major advances (P2-3).
- Page 2 Bottom of the page & Discussion paragraph 4.2. and 4.3 When exploiting the stop-signal task, bilateral deep brain stimulation (DBS) of STN has been repeatedly shown to improve reactive (e.g., Mirabella et al. 2012; van den Wildenberg et al. 2006, Swann et al. 2011, Mirabella et al. et al. 2012; van Wouwe et al. 2017) and proactive inhibition of upper limb movements (Mirabella et al. 2013). Notably, Mirabella et al. 2013 showed that bilateral stimulation of STN restores a planning strategy for reaching arm movement generation appropriate to the context in which Parkinson's patients are embedded, i.e., similar to those of healthy subjects. In Mirabella et al. (2013), proactive inhibition was evaluated by comparing the reaction times (RT) and the movement times (MT) of no-stop trials with those measured during the execution of the same movements in the context of a simple RT-task (go-only trial). In fact, it has been repeatedly shown (Mirabella et al. 2008) that when a subject performs a no-stop trial, its RT is lengthened, and its MT is shortened with respect to when he/she performs a go-only trial. This phenomenon represents an optimization of the motor strategy in the two different contexts, and thus, it has been named 'context effect' (Mirabella et al. 2008). Interestingly, as far as the laterality issue of the inhibitory network is concerned, unilateral STN DBS does not affect reactive or proactive inhibitory control of upper limb movements (Mancini et al., 2019). I suggest discussing this evidence because first, the assessment of the effect of proactive inhibition performed as described above minimize most if not all the confounds typical of more complex designs such as the conditional stop-signal (e.g., Aron et al. 2007) or stop-signal anticipation tasks (e.g., van Hulst et al. 2018). Under these conditions, the load on attentional and working memory is high. Thus, outcome measures cannot be easily ascribed to the effect of proactive strategies because of the concurrent cognitive demands on other executive functions. Second, DBS experiments, differently from recording techniques, allow establishing a causal link between the experimental manipulation and the behavioral outcomes.
We fully agree with Reviewer 2 and are grateful for these detailed and constructive comments. All these points are indeed central to various aspects of our rationale, and have been included in the revised version of our manuscript. These suggestions have inspired modifications not only in the introduction section (P2-3), but also in the methodological (P3-4) and in the discussion (P8-9) sections.
- Page 2 Third paragraph I believe that an original interpretation of inhibitory control's neural basis can be found in Mirabella (2014). In that paper, it has been proposed that acting and stopping represent functions emerging from specific interactions between largely overlapping brain regions, whose activity is intimately linked (directly or indirectly) to the evaluations of the pros and cons of an action. Such a mechanism would allow the brain to flexibly perform as different functions could be computed, exploiting the same components operating in various configurations. I suggest considering this hypothesis.
We agree with Reviewer 2. This general model has been included in the conceptual framework of our paper (P2). As also required in other terms by Reviewer 1, we have added a figure to visualize the working hypothesis within the global theoretical context (Fig.1). Reference to the conceptual framework of Mirabella (2014) has been provided.
- Page 3 Paragraph 2.2 In what sense you suggested participants 'to exert proactive inhibitory control'? Did you explicitly tell them to slow down the movements? This would represent a critical flaw. Subject automatically and largely unconsciously follow this strategy. If you provide explicit instruction, you can alter the experimental results. Please clarify.
No, absolutely not. On the contrary, we pushed subjects to react as fast as possible. Ensuring that responses are prepotent is indeed essential to guarantee the need for inhibition. We have clarified this point in the text, evoked the method used to make sure that subjects complied with this explicit recommendation (training sessions to calibrate fast RT distribution and feedback-based procedure to control RT distribution during recording sessions), and referred the reader to both: i) the detailed procedure described in the first paper using this dataset and to ii) former methodological studies demonstrating that the speeded simple reaction time task used here involves prepotent responses that elicit covert automatic motor activations (P3-4).
- Page 4 paragraph 2.2.4 I do not completely agree on the rationale of the contrast to uncover the neural underpinnings of a reactive no-selective inhibitory mechanism. In my opinion, you should compare the Go_control (certain condition) just with the Go (uncertain condition). This way, you would compare the same number of trials and two conditions where the visual stimuli are perfectly matched. I would suggest performing such analyses.
We acknowledge that the point raised by Reviewer 2 is critical. Yet, we have already provided answers to these legitimate questions in the 2017 paper based on the same dataset, as well as in others using the same behavioural design (e.g., Albares et al., Human Brain Mapping 2014). First, from a theoretical point of view, the reactive non-selective model of response inhibition (or global inhibition) assumes that both No-Go and Go trials presented in the same uncertain context trigger response inhibition even before they have been fully identified. There is therefore no a priori reason to discriminate these two stimuli. And, indeed, we always failed to find significant differences in early responses to these two stimuli. Second, the contrast [(no-go+go) - (go_control)] was balanced by weighting the go_control condition (x2) in order to compensate for the unequal number of trials in the red fixation point and green fixation point conditions. Information has been added in the text (P5).
- You found that reactive non-selective inhibition is underpinned by three basal ganglia in the left hemisphere, while the right hemisphere processed proactive inhibition. Maybe it would be worthy of discussing your findings in light of the influential idea suggesting that the right hemisphere is more involved in inhibitory control than the left hemisphere (e.g., Aron et al. 2014). This is a hotly debated question as several studies showed the opposite, i.e., they provide evidence showing that the inhibitory network subserving upper limb movements is distributed in both hemispheres (Klein et al., 2016). For instance, it has been shown that the left inferior frontal gyrus has a key role in inhibitory control (Swick et al. 2008). Furthermore, it has also been shown that bilateral (Mirabella et al. 2012; Mirabella et al. 2013) but not unilateral DBS (Mancini et al. 2019) restores reactive inhibitory control to a near-normal level. Besides, Mirabella et al. (2017) compared the inhibitory performance of right- and left-dominant Parkinson's patients (RPD and LPD, respectively) in the middle stage of the disease (H&Y-2 or -3), and they did not find any difference in either reactive or proactive inhibition between LPD and RPD patients, even though patients were impaired with respect to healthy controls. This finding has recently been confirmed, testing PD patients in the early stage of the disease when the disease is unilateral (Di Caprio et al. 2020). All in all, this evidence suggests that inhibitory control does not rely solely on the right hemisphere but on the cooperation between the two hemispheres. How do you frame your findings with respect to this topic?
We, again, fully agree with Reviewer 2. We have argued multiple times against the view that inhibitory control relies solely on a right lateralized circuit (e.g., Jaffard et al., 2008; Criaud and Boulinguez, 2013; Criaud et al., 2017). For the sake of clarity, this point has not been discussed in the first version of this paper, but we agree that this issue is not trivial. We have thus added some arguments in the discussion of the revised version (P8-9).
- CONCLUSIONS I would suggest including another consideration, i.e., understanding the interplay between proactive and reactive inhibition is a crucial step towards understanding disorder characterized by poor urge control (e.g., for a recent review, see Mirabella 2021).
We have extended the conclusion to include, as suggested, considerations about the clinical issues associated with the knowledge developed in our work (P10).
Minor points
All minor points have been considered carefully.
References
We are grateful to Reviewer 2 for providing these references. We have made good use of it.
